# ENABLE QUANTUM GRAPH NEURAL NETWORKS ON A SINGLE QUBIT

## ABSTRACT

In the present NISQ era, quantum machine learning encounters significant challenges in realizing practical implementations due to constrained resources. Recently, a novel single qubit methodology enabled the execution of numerous operations on a single qubit. However, the applicability of this approach to more intricate neural network architectures remains uncertain. In this study, we investigate graph data, which poses inherent challenges for machine learning due to its intricate structure. Through the utilization of quantum walk, we introduce a graph embedding approach to reduce network parameters, culminating in the design of a novel single-qubit Quantum Graph Neural Network (sQGNN) architecture. Our experimental assessments encompass both simulations and practical executions on quantum computers, revealing that the devised sQGNNs adeptly adapt to graphs of varying dimensions and compositions. Consequently, these single-qubit QGNNs make efficient use of limited qubit resources, paving the way for the realization of expansive quantum graph neural networks on resource-constrained Variational Quantum Circuits (VQCs). This breakthrough opens up new possibilities for practical applications within the current NISQ regime.

## 1 INTRODUCTION

The application of bio-inspired deep learning to machine learning has enabled numerous new applications (Min et al., 2017; Gligorijević et al., 2021). As information technology continues to advance and the density of information increases exponentially, innovative approaches to the complexity of machine learning have become imperative (Schuld et al., 2015). Quantum computing is a promising candidate with its superior computing power, thus gaining the attention of researchers in the field of artificial intelligence. Quantum algorithms have demonstrated an exponential acceleration advantage over classical solutions in high-dimensional data processing (Schuld et al., 2015). The combination of quantum computing and machine learning has resulted in the development of quantum machine learning, which has the potential to improve classical machine learning algorithms (Huang et al., 2021), yet faces hardware and software implementation challenges in the current NISQ era of quantum computing (Preskill, 2018).

Quantum computers have made impressive progress (Schuld et al., 2015), and some quantum machine learning methods have also been proposed (Benkner et al., 2021; Yang & Sun, 2022; Ovalle-Magallanes et al., 2022), though several major limitations remain in the NISQ era (Preskill, 2018). These include limited qubit numbers, with state-of-the-art devices ranging from 50 to 100 qubits, and architectural and programmability restrictions (Preskill, 2018). Additionally, high noise levels (Clerk et al., 2010) and the prospect of fault-tolerant quantum computers in years or decades (Preskill, 2018) prevent the realization of quantum supremacy (Preskill, 2012), that is, the acceleration of real-world applications. Therefore, the key question is how to make the most of current NISQ devices to achieve quantum advantages, taking into account the limited number of qubits, limited qubit connectivity, and coherent and incoherent errors which limit the depth of quantum circuits. Variational Quantum Algorithms (VQAs) (Cerezo et al., 2021) are widely used in quantum machine learning, but still have many limitations, such as trainability, accuracy, and efficiency, especially the "barren plateau" (McClean et al., 2018), and the details are shown in the appendix.

Graph-structured data has become a highly active research topic in the field of machine learning, with many applications in social networks, point clouds, molecular chemistry, and other areas (Min

et al., 2017; Xia et al., 2021). However, classical neural networks are designed to process data with regular structures in Euclidean space, and their efficiency is consequently reduced when dealing with complex graph data. Quantum computing has been suggested as a potential solution to address this computational complexity, as data in quantum machine learning is often represented in a high-dimensional Hilbert space by quantum states, which has been demonstrated to be beneficial for classification tasks (Schuld, 2019).

This paper proposes a novel method for implementing an entire quantum graph neural network on a single qubit (Pérez-Salinas et al., 2020; Easom-McCaldin et al., 2022), allowing for the analysis of complex graph data on resource-constrained NISQ VQCs, called Single-Qubit Quantum Graph Neural Network (sQGNN). Firstly, a graph embedding method based on discrete-time quantum walks is used to convert a graph data sample into a vector. Secondly, a new encoding scheme is used to easily encode the vectorized graph data onto quantum circuits, taking advantage of quantum computing to reduce the number of parameters required by classical neural networks, often to several tens from thousands and beyond. This drastically reduces the quantum volume occupancy. Thirdly, the uploaded quantum graph data is fed into a novel quantum graph neural network classifier that uses only a single qubit, which can be trained to fit the objective function through a variety of quantum rotation gates.

## 2 SINGLE-QUBIT GRAPH NEURAL NETWORK

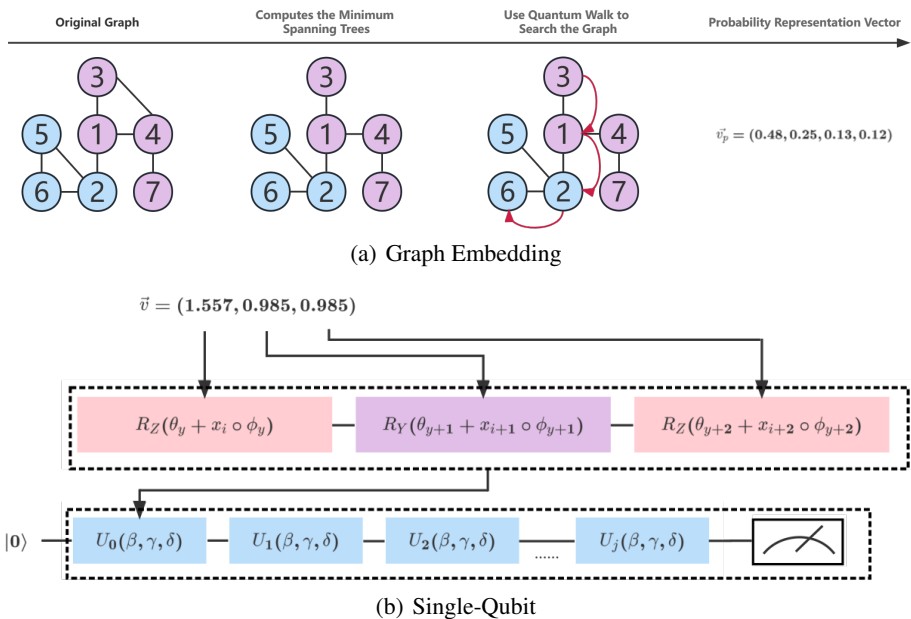

(a) Graph Embedding

(b) Single-Qubit

Figure 1: (a) The diagram of the graph embedding based on the quantum walk. (b) The Single-Qubit quantum circuit. The elements in the representation vector of the graph are in units of 3, and are encoded onto the qubits using a quantum rotation gate. Every three quantum rotation gates constitute a unitary operation, and the Single-Qubit method uses this unitary operation as the basic unit of encoding and parameter training.

Figure 1 illustrates the workflow of our proposed sQGNN method, which consists of three steps. First, the quantum walk graph embedding method is applied to convert raw graph data into a vector form that can be processed. Second, the processed graph representation vector is input into the single-qubit circuit in a predefined order of unitary operations, as depicted in Figure 1. This single-qubit method can encode all data with a single qubit, thus reducing the total number of qubits necessary for the NISQ era and effectively avoiding the "barren plateau" phenomenon (McClean et al., 2018) caused by excessive qubits in VQA. Finally, we measure the single-qubit circuit, calculate the fidelity and the loss of the model, and update the parameters of the quantum circuits to perform the training of the neural network. The training process provides us with the desired model.

Further details on these three steps can be found in the Methods section. Subsequently, experiments were conducted to evaluate our sQGNN method on several real-world datasets from the chemistry and biology domains.

In summary, our method has the following advantages: **Save quantum resources**. Our approach requires only one qubit, which is significant in the NISQ era of quantum computing resources. **Anti-noise**. Our model exhibits good noise immunity on simulated depolarization errors as well as real quantum computers. The good robustness of the quantum method itself can save the extra quantum resources required by the Quantum Error Correction (QEC) algorithm. **Avoid the "barren plateau"**. Our method only needs one qubit, effectively avoiding the "barren plateau". **The amount of parameters is small**. Our quantum approach has significantly fewer model parameters than the classical approach. **Extensibility**. Our quantum model can be combined with other quantum or classical structures to form a hybrid neural network. **Easy to deploy**. Our model requires only one qubit and can be easily deployed on real quantum computers.

## 2.1 QUANTUM WALK BASED GRAPH EMBEDDING

In order to embed a graph onto a single qubit, classical graph data needs to be processed. Our method is inspired by (Bai et al., 2020). When given a group of graphs, the method employs discrete-time quantum walks (Emms et al., 2009) to analyze them. Compared to classical walks, quantum walks possess remarkable features due to their dependence on unitary matrices instead of random ones. As they evolve, quantum walks create patterns of destructive interference, leading to improved graph representations (Emms et al., 2009). Additionally, quantum walks operate in the state space of directed edges rather than vertices, allowing for enhanced detection of the graph's structure. A coordinate wave function can be used to describe the particle position:

$$|\psi_p\rangle = \sum_x c_x |x\rangle \tag{1}$$

Where $|x\rangle$ is the wave function of particle in position $x$ and $c_x$ is the complex amplitude. So the probability of finding a particle at $x$ is:

$$p(x) = |c_x|^2 \tag{2}$$

The particle is in a superposition state of various positions during the process of walking. Considering the random walk of particles, we need to introduce the flipping operator $C$ and moving operator $S$:

$$C = H \otimes I_x \tag{3}$$

$$S = |0\rangle \langle 0| \otimes |x+1\rangle \langle x| + |1\rangle \langle 1| \otimes \sum_x |x-1\rangle \langle x| \tag{4}$$

Where $H$ is the Hadamard gate and $I_x$ is the identity matrix of wave function of $x$. Apply these two operators to the initial state $|\psi_0\rangle$ where the position is 0 and the state is upspin, the next state is:

$$|\psi_1\rangle = \frac{1+i}{2} |0\rangle| 1_x\rangle + \frac{1+i}{2} |0\rangle| -1_x\rangle \tag{5}$$

The state $|\psi_1\rangle$ is in superposition of position $x = -1$ and $x = 1$. This process can obtain a different probability distribution than the classical process.

To begin, we apply the commute time spanning tree (CTST) representation of the input graphs. Next, we execute simulations involving the evolution of a discrete-time quantum walk across the CTSTs. With access to the CTST representations of the graphs, we proceed to simulate the evolution of a discrete-time quantum walk on each of the trees, employing the Perron-Frobenius operator (Ren et al., 2011). Afterwards, we calculate the associated time-averaged density matrix for each quantum walk. This matrix captures the statistical set of quantum states resulting from the time evolution

of the quantum walk. Each element along the main diagonal of this matrix represents the time-averaged probability of the walk occupying an edge in the underlying graph. These probabilities effectively characterize the likelihood of the walk moving along specific arcs throughout the course of its evolution. Based on this, we can obtain a representation vector for each graph. The specific work embedding process is shown in Figure 1(a).

## 2.2 SINGLE-QUBIT BASED QUANTUM ENCODING OF GRAPH DATA

Data encoding for many Machine Learning (ML) tasks is often presented as column vectors of classical data. Single-qubit encoding, introduced in (Pérez-Salinas et al., 2020), is a strategy for encoding a vector of classical data into a characteristic Hilbert space through a series of single operations acting on each input data dimension, applied to a single qubit as shown in Figure 1(b). The unitary operation of single-qubit encoding can be expressed by the following formula:

$$U = R_Z(\beta) R_Y(\gamma) R_Z(\delta) = e^{i\beta\sigma_z} e^{i\gamma\sigma_y} e^{i\delta\sigma_z} \tag{6}$$

Where $\sigma$ is the Pauli matrix. With Euler angles $\beta$, $\gamma$, $\delta \in \mathbb{R}$ that define the extent of each rotation ($R$) around the Z, Y and Z axes respectively. Within this method of encoding, these Euler angles are parameterized further and defined as:

$$\beta = \theta_i + x_i \cdot \phi_i \tag{7}$$

$$\gamma = \theta_{i+1} + x_{i+1} \cdot \phi_{i+1} \tag{8}$$

$$\delta = \theta_{i+2} + x_{i+2} \cdot \phi_{i+2} \tag{9}$$

Where $\theta_i$ and $\phi_i$ are trainable weight parameters assigned to $x_i$, the value of the input vector $x$ at dimension $i$. Therefore, the extent of rotation $\beta$, $\gamma$, $\delta$ is with respect to the weighted value of the input. Combining three rotation gates into a unitary operation, there is:

$$U\left(\vec{\omega}\right) = e^{i\vec{\omega}\cdot\vec{\sigma}} \tag{10}$$

Where $\vec{\omega} = (\omega(\beta), \omega(\gamma), \omega(\delta))$. And respectively:

$$\omega(\beta) = c\left(\sqrt{1 - cos^2 c}\right)^{-1} \sin\left(\frac{\gamma - \delta}{2}\right) \sin\left(\frac{\beta}{2}\right) \tag{11}$$

$$\omega(\gamma) = c\left(\sqrt{1 - cos^2 c}\right)^{-1} \cos\left(\frac{\gamma - \delta}{2}\right) \sin\left(\frac{\beta}{2}\right) \tag{12}$$

$$\omega(\delta) = c\left(\sqrt{1 - cos^2 c}\right)^{-1} \sin\left(\frac{\gamma + \delta}{2}\right) \cos\left(\frac{\beta}{2}\right) \tag{13}$$

Where $\cos c = \cos\left(\frac{\gamma + \delta}{2}\right) \cos\left(\frac{\beta}{2}\right)$. The single-qubit encoding method can be employed to encode up to three input dimensions per unitary operation. The input vector is thus cycled through in order to encode three-dimensional values until the entire input has been encoded. This method can be flexibly implemented on quantum circuits that process data of different structures and can increase the data capacity per qubit.

For classic models, there is Universal Approximation Theorem (UAT) (Hornik, 1991) to support its approximation capabilities. Similarly, for quantum models, UAT can be used to demonstrate approximation capabilities. According to (Pérez-Salinas et al., 2021), a quantum analogue can be constructed on the basis of UAT. Let $f$ and $\varrho$ be a pair of functions, with $f \in \mathbb{R}^m \to [0, 1]$ and $\varrho \in \mathbb{R}^m \to [0, 2\pi)$, there is:

$$\left| f\left(\vec{x}\right) e^{i\varrho(\vec{x})} - \left\langle 1 \left| \prod_{i=1}^{N} U\left(\vec{x}, \vec{\theta}_i, \vec{\phi}_i\right) \right| 0 \right\rangle \right| < \epsilon \tag{14}$$

Where $\epsilon > 0$. Based on this quantum UAT, it can be considered that the Single-Qubit method is able to approximate the functions.

## 2.3 Measurement and Loss Calculation in Our Single-Qubit GNN

In the measurement phase, we observe the quantum circuit to obtain its final state. Due to the relatively special step of measurement in quantum circuits compared with classical methods, we can use the characteristics of measurements to construct an evaluation method similar to entropy. As opposed to traditional direct measurement of the qubit state, we adopt a fidelity-based measurement method for classification tasks. In quantum information theory, fidelity is a measure of the "closeness" of two quantum states. It represents the probability of a state being identified as another state by a measure. Fidelity is defined as follows:

$$F\left(\rho, \sigma\right) = \left( tr\sqrt{\sqrt{\rho}\sigma\sqrt{\rho}} \right)^2 \tag{15}$$

Where $\rho$ and $\sigma$ are two quantum state. The goal of this method is to minimize the fidelity between a set of data encodings and their respective target states. For the Single-Qubit method, we only need to measure once for each graph data. For binary classification, we assign each image from the set size D with class values in 0,1 to the target states $|0\rangle$ or $|1\rangle$. This approach can be applied to any number of classes, provided that the target states are maximally distinguishable. Fidelity $F$ is a metric for the similarity or proximity between two quantum states, in which $0 \leq F \leq 1$. The higher the fidelity, the more similar the states are in the direction. The highest class fidelity value given is then considered the classification result. The formula of fidelity is shown as follows:

$$F\left(\vec{x}, \vec{\theta}, \vec{\phi}\right) = \left| \left\langle \vec{\psi}_l \middle| \psi_{output}\left(\vec{x}_i, \vec{\theta}, \vec{\phi}\right) \right\rangle \right|^2 \tag{16}$$

The role of the loss function is to quantify the discrepancy between the predicted output of the model and the true value. In our model, fidelity is used to measure the similarity between the results and the labels, and hence the loss function is also designed accordingly. The formula is as follows:

$$\mathcal{L}\left(\vec{x}, \vec{\theta}, \vec{\phi}\right) = \sum_{i=1}^{M} \left( 1 - F\left(\vec{x}, \vec{\theta}, \vec{\phi}\right) \right) \tag{17}$$

Where $\vec{\psi}_l$ is the correct label state of the data point.

Having obtained the loss, we can use the optimizer to maximize the sum of the fidelity of all data points and find the best weight for classification, the parameters $\theta_i$ and $\phi_i$ in the unitary operation above. The optimization strategy we use is Quantum Natural Gradient, based on (Stokes et al., 2020). A distinctive feature of the quantum state space is its possession of an intrinsic metric tensor known as the Fubini-Study metric tensor. By capitalizing on this property, we can develop quantum natural gradient descent:

$$\omega_{t+1} = \omega_t - \eta g^+\left(\omega_t\right) \nabla \mathcal{L}(\omega) \tag{18}$$

Where $g^+$ is the pseudo-inverse of the Fubini-Study metric tensor. The following is a variational quantum circuit:

$$U(\omega)|\psi_0\rangle = V_l\left(\omega_l\right) V_{l-1}\left(\omega_{l-1}\right) \cdots V_0\left(\omega_0\right)|\psi_0\rangle \tag{19}$$

Where $V_l\left(\omega_l\right)$ are layers of quantum gates with parameters. $\omega$ represents the parameters, including the above mentioned $\beta$, $\gamma$ and $\delta$. Considering that only rotation gates are used in the Single-Qubit method, these quantum gates can be transformed into the following form:

$$X\left(\omega_i^{(l)}\right) = e^{i\omega_i^{(l)}K_i^{(l)}}K_i^{(l)} \tag{20}$$

The block-diagonal submatrix of the Fubini-Study tensor is:

$$g_{ij}^{(l)} = \langle\psi_{l-1}|K_iK_j|\psi_{l-1}\rangle - \langle\psi_{l-1}|K_i|\psi_{l-1}\rangle\langle\psi_{l-1}|K_j|\psi_{l-1}\rangle \tag{21}$$

$$|\psi_{l-1}\rangle = V_{l-1}\left(\omega_{l-1}\right)\cdots V_0\left(\omega_0\right)|\psi_0\rangle \tag{22}$$

Based on this, through quantum backpropagation, we can train the quantum model at a faster speed.

## 3 EXPERIMENTS

We chose the MUTAG dataset and PTC series dataset as experimental data. The MUTAG dataset contains 188 nitro compounds. The graph data belongs to the isomer graph. The full name of PTC is Predictive Toxicology Challenge, which is used to develop advanced SAR technology predictive toxicology models. According to the experimental rodent species, there are a total of 4 datasets: PTC_FM, PTC_FR, PTC_MM, and PTC_MR. The specific parameters of the dataset are shown in the table below. We use PennyLane (Bergholm et al., 2018) and PyTorch (Paszke et al., 2019) to perform experiments. The quantum computer we use is the ibm_manila node provided by IBM (IBM, 2023). The number of qubits available to this quantum computer is 5, the Quantum volume is 32, it can perform 2800 Circuit layer operations per second, and the processor model is Falcon r5.11L.

### 3.1 EXPERIMENTS IN THE SIMULATED QUANTUM ENVIRONMENT

We configured two simulation setups for our sQGNN models: an ideal quantum environment and a noisy environment involving noise interference in the simulated NISQ devices. To evaluate the robustness of the sQGNN model, we conducted multiple experiments under both simulated environments and tested the classic models with edGNN(Jaume et al., 2019), R-GCN(Schlichtkrull et al., 2018), GIN(Xu et al., 2018), RW-GNN(Nikolentzos & Vazirgiannis, 2020), TOGL(Horn et al., 2022) and sQGNN. The tested quantum models include GBS(Schuld et al., 2020), QJSK and QJSKT(Bai et al., 2017). To verify the Single-Qubit method, we set up a control model composed of two quantum bits. The two qubits of the sQGNN-Dual model are the same. The input data is encoded once in the two qubits, and there is a quantum entanglement between the two qubits.

Table 1: The average accuracy of the models over different real-world graph datasets.

| Model | MUTAG | PTC_FM | PTC_FR | PTC_MM | PTC_MR |
|---|---|---|---|---|---|
| edGNN | 86.9±1.0 | 59.8±1.5 | 65.7±1.3 | 64.4±0.8 | 56.3±1.9 |
| R-GCN | 81.5±2.1 | 60.7±1.7 | **65.8±0.6** | 64.7±1.7 | 58.2±1.7 |
| GIN | 85.4±3.5 | 64.4±6.7 | 65.1±5.3 | **64.8±5.4** | **64.6±7.0** |
| RW-GNN | 88.3±4.1 | 60.9±2.7 | 63.1±1.3 | 63.2±1.4 | 57.1±1.4 |
| TOGL | 87.2±3.8 | **64.9±4.6** | 64.8±3.8 | 63.1±4.1 | 60.3±4.7 |
| GBS | 86.4±0.3 | 53.8±1.0 | - | - | - |
| QJSK | 83.4±0.5 | - | - | - | 58.2±0.8 |
| QJSKT | 81.6±0.5 | - | - | - | 57.4±0.4 |
| SQGNN | **87.3±4.8** | 66.2±7.3 | 66.9±5.5 | 65.9±4.2 | 65.9±5.5 |
| SQGNN-Dual | 87.2±4.2 | 66.2±6.9 | 66.4±4.8 | 66.0±4.1 | 65.9±5.1 |

**Ideal simulated environment**. Table 1 presents the test results of the sQGNN model in an ideal simulated quantum environment. The best result is underlined and the second result is bolded. The results indicated that the sQGNN model achieved the best performance on the four datasets of PTC

Table 2: The average accuracy of the models with depolarizing error.

| depolarization probability | MUTAG | PTC_FM | PTC_FR | PTC_MM | PTC_MR |
|---|---|---|---|---|---|
| **0.001** | 87.3±3.3 | 65.0±7.2 | 66.5±9.1 | 65.3±4.2 | 65.8±4.4 |
| **0.01** | 86.7±4.9 | 65.1±8.6 | 66.1±5.6 | 65.7±4.5 | 65.8±5.1 |
| **0.1** | 84.8±6.4 | 64.0±7.7 | 65.0±8.4 | 63.7±4.8 | 64.5±4.5 |

datasets(Helma et al., 2001). As a research (Easom-Mccaldin et al., 2021), increasing the number of processing layers (i.e., unitary operations) of the single-qubit method improves the model's performance, especially when the number of unitary operations is below three. This could be attributed to the relatively low number of feature elements in the graph representation vector of MUTAG data.

The comparison of the results in the ideal quantum environment revealed that the overall accuracy of our sQGNN model was superior. Moreover, the sQGNN method has the advantage of a compact model structure. Classical graph neural networks tend to be complex in terms of the model structure due to the complexity of graph data itself. By contrast, our model, which employs a graph embedding method without hyperparameters and Single-Qubits QGNN quantum circuits, obtains a huge advantage in the total number of parameters. For instance, in the MUTAG dataset, our model requires only 12 parameters, whereas a classical neural network such as GIN has 13,000 parameters.

The performance of sQGNN and sQGNN-Dual models did not show a significant difference. The standard deviation of the sQGNN-Dual model is relatively small, but this is foreseeable, because the input data of the sQGNN-Dual model has been entered once more once, and the number of parameters increases. In the task set in the experiment, the number of quits increased significantly, but the quantum resources used were doubled.

**Noisy simulated environment**. To verify the robustness of sQGNN against noise when running on NISQ devices, we train and test the models in a noisy simulated environment. As shown in Table 3, the depolarization error was set as the noise in the environment for the sQGNN model. The depolarization error applies a random Pauli (e.g. X, Y, Z) to each qubit in a group of qubits and is described by the following formula:

$$E(\rho) = (1 - \varepsilon)\,\rho + \varepsilon Tr[\rho]\frac{I}{2^n} \tag{23}$$

Where $\varepsilon$ is the depolarizing error param, $n$ is the number of qubits for the error channel and $I$ is the Pauli matrix.

We set three levels of the depolarization parameter to assess the tolerance of sQGNN to depolarization errors. The test results are shown in Table 3 and reveal that sQGNN is almost unaffected by depolarization errors at 0.001 and 0.01, and slightly affected at 0.1. Despite the functional limitations of quantum devices in the NISQ era, the influence of noise is unavoidable, resulting in significant degradation of performance on real quantum computers. However, the results of this experiment show that the sQGNN model can stably maintain good performance in the presence of noise.

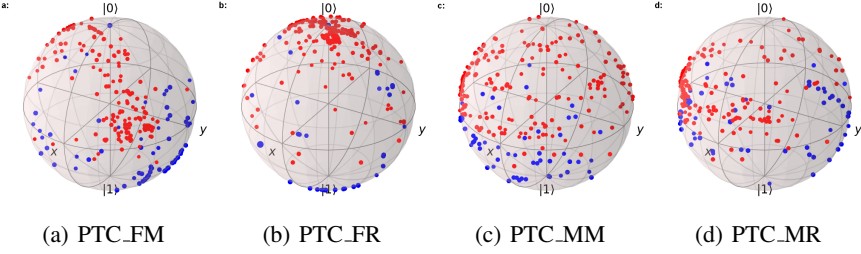

(a) PTC_FM      (b) PTC_FR      (c) PTC_MM      (d) PTC_MR

Figure 2: This figure is a demonstration of all graph data of the PTC dataset in Hilbert space. The red dot is class 0 and the blue dot is class 1.

Figure 2 displays the distribution of graphs from the PTC_FM dataset in Hilbert space on the surface of Bloch spheres. Red points correspond to Class 0 and blue points to Class 1. Class 0 is mainly concentrated in the upper hemisphere, while Class 1 is mainly concentrated in the lower hemisphere. This suggests that the model's classification ability for Class 0 is stronger than for Class 1, with some outliers distributed in the upper hemisphere. This may be due to the initial state of the quantum circuit being set to $|0\rangle$ or the structure of the graph data.

## 3.2 REAL QUANTUM DEVICE TEST EXPERIMENT

We evaluated the performance of our single-qubit sQGNN model on the MUTAG dataset using a real quantum computer provided by IBM. The model was deployed in a simulated quantum environment to an IBM online quantum computer, facilitated by IBM Quantum Lab.

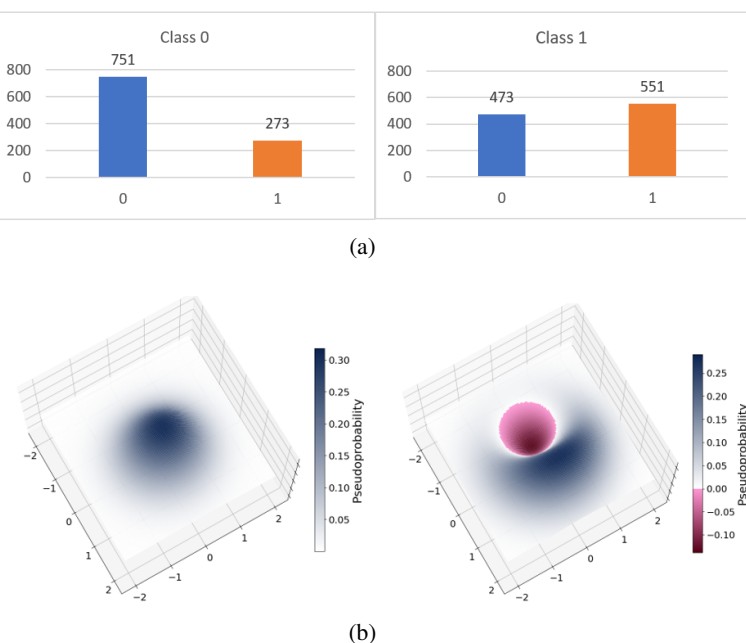

Figure 3: (a) Results of testing the effects of the model on IBM's quantum computer, ibm_manila. Here are the measurements of two graphs with classes 0 and 1 in the test data. The ordinate represents the number of occurrences of a particular state in the measurement. (b) Visualization of the Wigner quasi-probability function for Class 0 and 1. Our model discriminates between the two classes clearly on a quantum computer.

Figure 3 demonstrates the test results of two samples on the real IBM quantum computer and the corresponding simulated quantum environment. The accuracy obtained on the MUTAG dataset was 88.89%, with no degradation in accuracy compared to the simulation environment. Furthermore, the measurement results of $|0\rangle$ and $|1\rangle$ are shown in Figure 3(a), leading to the observation that Class 0 is better classified than Class 1. This phenomenon can be attributed to the fact that the initial state is $|0\rangle$ and the difference in the graph data structure of the two classes, which is consistent with the results obtained in the simulated environment in Figure 2.

Figure 3(b) presents the visual image of the Wigner quasi-probabilistic equations (Hiley, 2004), which relate the wave function to a probability distribution in phase space. The results in the figure illustrate that the model can effectively differentiate between the two classes, and is less affected by noise generated from NISQ devices. Generally, the initial state of available quantum computers is set to $|0\rangle$. If the initial state is set to $|1\rangle$, the outcome may differ. Due to the physical limitations of the system, the initial state of $|0\rangle$ may remain the norm of quantum computers for a period of time, thus requiring the optimization of the algorithm to address this issue.

## 4 DISCUSSION

In the NISQ era, the implementation of quantum algorithms on noisy devices is limited by unavoidable device noise. To address this issue, single-qubit method has been proposed as an efficient solution for exploiting the limited qubits available in the NISQ era. Their simple structure and high efficiency suggest that our sQGNNs could be extended to other tasks in the near future. This method has the potential to economize quantum resources in the NISQ era, where quantum computing resources are scarce.

Nevertheless, experimental evidence indicates that the single-qubit method utilized in our proposed model demonstrates resilience to noise in both simulated and real-world quantum computing settings. Although the Quantum Error Correction (QEC) algorithm is a common approach to mitigate errors arising from device noise, its implementation demands supplementary resources. In this regard, the inherent robustness of our algorithm offers a means to conserve valuable resources, alleviating the need for costly QEC mechanisms in the NISQ era.

It is obvious that, due to the inherent coherence of quantum systems, the single-qubit method theoretically makes sQGNN less prone to quantum decoherence. However, recent theoretical analysis on single qubit methods (Yu et al., 2022) has raised questions about whether such implementations might limit the model's expressivity of multivariate functions and weaken its robustness. To address these concerns, we conducted experiments that included a dual-qubit implementation of sQGNN for comparison. The results, as displayed in Table 1, indicate that sQGNN demonstrated slightly better accuracy than sQGNN-Dual. This finding suggests that there is no significant difference in performance whether one or two qubits are used, thus alleviating concerns regarding the efficacy of single-qubit methods.

Given the efficient utilization of the sQGNN method with a single qubit, we are able to keep the total number of qubits at a minimal level, thus circumventing the training difficulties of the "barren plateau" induced by VQC. While some studies (McClean et al., 2018; Grant et al., 2019; Stokes et al., 2020) have proposed methods to ameliorate the "barren plateau", a fundamental solution to this issue has yet to be discovered. In the current state of quantum machine learning methods applying VQA, the "barren plateau" may persist for a while. The Single-Qubit method can be an effective way at this stage. More discussion can be found in Appendix A.

## 5 CONCLUSION

In conclusion, to address the challenges on the constraints of limited quantum resources facing the current VQCs of NISQ era, we successfully leveraged single qubit strategy and developed a novel quantum graph neural network architect on a single qubit by integrating a quantum walk graph embedding method with sQGNN quantum circuits that can successfuly reduce the required parameters from over 10k to only 12 parameters (for MUTAG dataset). Our new method forms a concise structure suitable for deployment on quantum devices during the NISQ era. Our experimental results underscore the model's robustness on both quantum computing simulators and real quantum computers, showing its great potential on overcoming resource bottlenecks, achieving robust performance, withstanding quantum noises, mitigating the "barren plateau" issues, etc., showing a groundbreaking step toward harnessing quantum machine learning for real-world applications within the current NISQ era, where quantum hardware resources are constrained.

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

# A    BARREN PLATEAU

Using variational quantum circuits and convolutions on quantum neural networks, we propose a new variational circuit-based quantum convolution model in this study. To examine different experimental results and carry out in-depth analysis, we run different experiments on different datasets. We use rotation gates to convert classical data into quantum data, and there are three rotation gates on each qubit so that the wave function can better fit the objective function. Our quantum convolution kernel can be combined with various classical models, which can bring quantum advantages to classical methods. We use quantum entanglement to emphasize the correlation between data in the region. Compared with some previous quantum models, these advantages make our method enables Quantum Biometrics. Our approach has great potential for data relevance and privacy. The quantum circuit in our method maintains a relatively simple structure while being easy to expand, and can effectively avoid the problem of gradient disappearance caused by the "barren plateau". The objective function gradient's mean value is zero for a large type of quantum circuit, and the probability that any particular instance of a given random circuit deviates from this mean value by a small constant is proportional to the qubit number. The value of any reasonably smooth function tends to its mean with exponential probability when the space's measure is concentrated in this way, meaning that the gradient is zero over a wide range of quantum space. This is the "barren plateau".

There is a variational quantum circuit (VQC):

$$U\left(\vec{\theta}\right) = U\left(\vec{\theta}_1, \cdots, \vec{\theta}_L\right) = \prod_{l=1}^{L} U_l\left(\vec{\theta}_l\right) P_l \tag{24}$$

Where $\zeta$ is the angle of the unitary operator, $P$ is the unitary operator with no angle parameter (i.e. Pauli X). In VQC, the expected value of the quantum circuit needs to be obtained. Consider an objective function $E(\zeta)$ expressed as the expectation value over some Hermitian operator $H$:

$$E\left(\vec{\theta}\right) = \langle 0| U\left(\vec{\theta}\right)^{\dagger} H U\left(\vec{\theta}\right) |0\rangle \tag{25}$$

The gradient of the objective function:

$$\partial_k E \equiv \frac{\partial E\left(\vec{\theta}\right)}{\partial \theta_k} = i \langle 0| U_-^{\dagger} \left[V_k, U_+^{\dagger} H U_+\right] U_- |0\rangle \tag{26}$$

$V$ is a a Hermitian operator. $U_+$ and $U_-$ are two circuits, both match the Haar distribution up to the second moment, and the circuits are independent. The average value of the gradient can then be expressed as:

$$\langle \partial_k E \rangle = \int dU p(U) \partial_k \langle 0| U\left(\vec{\theta}\right)^{\dagger} H U\left(\vec{\theta}\right) |0\rangle \tag{27}$$

Where $p(U)$ is the probability distribution function of $U$. In different cases, its variance is:

$$Var\left[\partial_k E\right] \approx \begin{cases} -\frac{Tr\left(\rho^2\right)}{(2^{2n}-1)} Tr\left\langle \left[V, u^{\dagger} H u\right]^2 \right\rangle_{U_+} \\ -\frac{Tr\left(H^2\right)}{(2^{2n}-1)} Tr\left\langle \left[V, u^{\dagger} \rho u\right]^2 \right\rangle_{U_-} \\ \frac{1}{2^{(3n-1)}} Tr\left(H^2\right) Tr\left(\rho^2\right) Tr\left(V^2\right) \end{cases} \tag{28}$$

Among them, the number of qubits is $n$. This means that when the number of qubits is large, in most cases, the gradient of the cost function approaches 0, that is, any training method based on VQA will not be able to make the cost function converge. In general, for a VQC, if the number of qubits in the quantum circuit exceeds 10, the VQC can hardly converge. Our model limits the number of qubits to 4 to avoid the "barren plateau"

