# OpenReview forum: "Enable Quantum Graph Neural Networks on a Single Qubits"
_ICLR.cc/2024/Conference — ICLR 2024 Conference Withdrawn Submission_

### Official Review · Reviewer_w9tL · 2023-10-23

**Soundness:** 2 fair
**Presentation:** 2 fair
**Contribution:** 2 fair
**Rating:** 3
**Confidence:** 3

**Summary:**

The paper introduces a quantum walk based graph embedding method and a single-qubit based quantum encoding scheme to process complex graph data, and evaluates the performance and robustness of the sQGNN model on two graph datasets and compares it with classical and quantum baselines. Moreover, the paper demonstrates that the sQGNN model can achieve comparable or better accuracy with significantly fewer parameters and qubits, and can withstand noise interference from NISQ devices.

**Strengths:**

- The paper optimizes the quantum circuit using the quantum natural gradient method, which can achieve faster convergence for quantum machine learning problems.

- The paper has the experiments conducted in a noisy simulated environment and a real quantum device.

**Weaknesses:**

- **The innovation of this paper is limited.** The proposed method mainly contains two modules, a quantum walk graph embedding and a single-qubit based quantum encoding, These two modules are the most commonly used in quantum methods.

- **The description of the proposed method is unclear.**  For example, the relationship between the size of the graph and the dimension of the representation vector of the final output graph is not described in this paper. Although Figure 1 gives a graphical example, it shows that the quantum walk module outputs a four-dimensional vector $v_p$,  but a three-dimensional vector $v$ is then encoded into the single qubit quantum circuit, where the relationship between $v_p$ and $v$ is not explained.

- **The application of the proposed method is limited.** The use of quantum walking means that the algorithm cannot take into account the feature of nodes or edges, making it impossible to apply the algorithm to a wider range of tasks. On the other hand, encoding all the information in a single qubit inevitably makes it impossible to deal with larger scale problems.

- **The comparison of related work is missing.** The proposed method is defined as a quantum graph neural network (QGNN), but no related method for QGNN based on the quantum circuit is discussed or contrasted, for example [1,2,3].

References:

[1] Guillaume Verdon, Trevor McCourt, Enxhell Luzhnica, Vikash Singh, Stefan Leichenauer, and Jack Hidary. Quantum graph neural networks. arXiv preprint arXiv:1909.12264, 2019.

[2] Jin Zheng, Qing Gao, and Yanxuan Lu. Quantum graph convolutional neural networks. In ¨ Proceedings of the Chinese Control Conference, pp. 6335–6340. IEEE, 2021

[3] Xing Ai, Zhihong Zhang, Luzhe Sun, Junchi Yan, and Edwin Hancock. Towards quantum graph neural networks: an ego-graph learning approach. arXiv preprint arXiv:2201.05158, 2022.

**Questions:**

1. What are the benefits of using the commute time spanning tree (CTST) to represent the input graphs?  Compared with the quantum walk on the original graph, how much of a performance boost can the proposed approach provide?

2. How to ensure the scalability of the model? As the embedding vector of the graph increases with the size of the graph, the information encoded on a single qubit also increases. Although a single qubit can approximate functions, how do you ensure that a single qubit can efficiently encode enough features?

3. How does this method perform compared to QGNN based on quantum circuits[1,2,3]?

---

### Official Review · Reviewer_uTG1 · 2023-10-23

**Soundness:** 3 good
**Presentation:** 3 good
**Contribution:** 2 fair
**Rating:** 3
**Confidence:** 2

**Summary:**

The authors propose a novel method for implementing a quantum graph neural network on a single qubit, which can analyze complex graph data on resource-constrained NISQ devices. The method consists of three steps: graph embedding based on quantum walks, single-qubit quantum encoding of graph data, and measurement and loss calculation using fidelity. The authors conduct experiments on several real-world graph datasets from chemistry and biology domains, and show that their method achieves competitive or superior performance to classical and quantum baselines. They also test their method on a real quantum computer provided by IBM and demonstrate its feasibility and effectiveness.

**Strengths:**

1. The SQGNN designed by the authors can save quantum resources by using only one qubit, which is significant in the NISQ era of quantum computing, which can be easily deployed on real quantum computers and combined with other quantum or classical structures.

2. It can resist noise and avoid the “barren plateau” phenomenon that affects variational quantum algorithms, which is helpful for applying modern quantum variational techniques for practical applications.

**Weaknesses:**

1. From the reviewer's perspective, the idea of mentioning and using the single-qubit quantum neural network structure in the paper is not originally proposed by the authors. The same concept has already been introduced in Paper [1], and the power and the limitations of the single-qubit quantum neural network has been investigated in [2]. Therefore, the reviewers have a negative attitude towards the contribution of the authors of this paper, both in terms of theory and experimentation.

2. The quantum neural network used in this paper is general (i.e., it does not specifically adapt its network structure for graph-structured data). The entire model's capability to handle graph-structured data actually comes from the module preceding the quantum neural network - graph embedding. The graph embedding, proposed by the authors based on quantum random walks, is also well-defined, such as [3,4]. The reviewers, however, have not identified any new contribution from the authors to existing technology.

3. The reviewer notes that the authors did not report the numerical values of the best performance for some baselines in the experimental results table (Table 1). Reviewers would like the authors to provide information on the hyperparameters, the number of parameters, training time, etc., for the sqgnn, sqgnn-dual, and baseline models. This information would facilitate a more in-depth comparison and reproduction for the readers.

4. The reviewer believes that the experimental section lacks crucial comparative experiments. Firstly, the graph embedding technique proposed by the authors based on random walk requires comparison with other graph embedding techniques, including DeepWalk, Node2Vec, matrix factorization, graph neural networks, etc. Secondly, the single-qubit QNN used by the authors also needs to be compared with other paradigms of quantum neural networks, such as hardware-efficient models and quantum neural networks specifically designed for graph-structured data.


[1] Pérez-Salinas A, Cervera-Lierta A, Gil-Fuster E, et al. Data re-uploading for a universal quantum classifier[J]. Quantum, 2020, 4: 226.

[2] Yu Z, Yao H, Li M, et al. Power and limitations of single-qubit native quantum neural networks[J]. Advances in Neural Information Processing Systems, 2022, 35: 27810-27823.

[3] Kempe J. Quantum random walks: an introductory overview[J]. Contemporary Physics, 2003, 44(4): 307-327.

[4] Ye X, Yan G, Yan J. VQNE: Variational Quantum Network Embedding with Application to Network Alignment[C]//Proceedings of the 29th ACM SIGKDD Conference on Knowledge Discovery and Data Mining. 2023: 3105-3115.

**Questions:**

Please see the weaknesses.

---

### Official Review · Reviewer_KmxC · 2023-10-31

**Soundness:** 3 good
**Presentation:** 2 fair
**Contribution:** 3 good
**Rating:** 5
**Confidence:** 4

**Summary:**

The authors introduce a new method for analysing complex graph data using single qubit quantum graph neural network (SQGNN) architecture.  The method uses discrete-time quantum walk to embed the graph which is later encoded using a single qubit thereby reducing the required number of parameters from thousands to twelve (for MUTAG dataset) though both methods were independently studied in the literature. The proposed method is illustrated using numerical simulations on MUTAG and PTC series dataset as well as executing on quantum computer. The robustness of sQGNN method was demonstrated by simulating in noisy environment using depolarization error. The method is also verified using sQGNN-Dual where two qubits were used for encoding. The method reduces the resources required as well as overcomes the difficulties of the barren plateau.

**Strengths:**

The paper introduces new method using quantum neural networks by combining the existing ideas to efficiently classify graph datasets.  The method uses only single qubit thereby reducing the quantum resources required as well as mitigating the barren plateau issues. The method is also demonstrated to show robustness against noises. The quantum features also enable the method to reduce the parameters required from thousand to twelve.

**Weaknesses:**

Section 2 which describes the method is not written clearly and is hard to follow the steps leading to embedding of graph. Also, there is lack of continuity and proper explanation of variables used in equations in section 2.3. Also, from the experiment results shown in figure 3, the method is doubtful as the success probability is marginal for Class 1. There are a few other issues that is raised in the questions section below.

**Questions:**

1.	Why there is a difference in success probability for Class 0 and 1? It seems that the QGNN is not able to change the initial state.
2.	Why the simulations in ideal simulated environment as shown in table 1 has lesser fidelity than using noisy real quantum computers? Providing a figure similar to figure 3 for ideal simulated environment is good for comparison.
3.	The error bars for the results seems very high (in table 1 and 2). Is there a way to confine the errors?
4.	Using two qubits for encoding seems to worsen the training as shown in table 1. Also, the explanation given for the reduction in standard deviation of dual model is not clear as increasing the number of qubits usually tend to increase the error rates.
5.	The literature about existing methods and its pros and cons compared to the new method is missing.

---

### Official Review · Reviewer_5o3F · 2023-11-01

**Soundness:** 2 fair
**Presentation:** 2 fair
**Contribution:** 2 fair
**Rating:** 3
**Confidence:** 5

**Summary:**

This paper presents a method of compressing the neural graph networks to a single qubit operations and shows experiments on the performance of the network. Paper is relatively well written and is clear enough to be understandable. The paper first presents a transformation of graph neural networks by the usage of finding a spanning tree and then employing the quantum random walk into a set of parameters that can be represented as a single vector. Then this vector is represented as a set of rotations in a learnable framework of VQA. The paper provides to some extent experimental verification.

**Strengths:**

The method of embedding using the random quantum walk. The approach is quite nice and interesting. It is a pity there are no experiments as to evaluate the complexity of the whole approach. As the main novelty this is a good result.

**Weaknesses:**

Lack of evaluation of the proposed method. As the authors say we are in NISQ but in general we should be target beyond NISQ. NISQ is a transitive period of intermediary quality technology that is in fact usable only experimentally. Therefore the main contribution, the single qubit embedding of the graph network is not evaluated enough. What is the computational cost of it?

In general there is not enough explanation about the principles of the working of the method in general.

This is even more crucial because the authors use a standard optimizer, that is the VQA. Therefore there should be a link explaining the difficulty of learning the graph networks as a function of the embedding not as a function of the VQA.

Minor problems: Table 2 is not cited. In addition, according to the authors, the resulting model has a smaller number of parameters. So by default using a mathematical model of noise it will be less noisy than model with more parameters.

Finally a comment on the evaluation. Considering that the authors were able to get 12 parameters in a problem, perhaps the method would be interesting frmo the optimization of existing neural quantum models .

**Questions:**

How does the learning accuracy depends from the maximum spanning tree of a graph in terms of speed.

What is the complexity in the number of gates and passes through the original graph?